# Study of citizen satisfaction and loyalty in the urban area of Guayaquil: Perspective of the quality of public services applying structural equations

**María Salomé Ochoa Rico[1]☯‡, Arnaldo Vergara-Romero[2]☯‡*, José Fernando Romero Subia[1]☯‡, Juan Antonio Jimber del Río[1]☯‡**

1 Department Agricultural Economics, Finance, and Accounting, University of Cordoba, Córdoba, Spain,
2 Department of Research, Ecotec University, Samborondon, Eucador

☯ These authors contributed equally to this work.
‡ MSOR, AVR, JFRS and JAJR also contributed equally to this work.
* avergarar@ecotec.edu.ec

**Data Availability Statement:** Data in the University of Córdoba database: http://hdl.handle.net/10396/22365.

## Abstract

This study investigates the satisfaction and adequacy of citizens through the expected quality and perceived quality in the areas of planning and territorial viability, experience in the provision of municipal services and citizen experience in environmental issues, in order to provide tools for territorial decision making for the citizens' well-being. In our research PLS software is used for the analysis of hypotheses. A questionnaire was delivered to a sample of 521 citizens, representing the spectrum of the population, and the statistical study of the responses yielded results on citizen satisfaction and loyalty. Our research includes the study of moderating effects on the causal ratio of perceived value and satisfaction in territorial planning and viability, the perceived quality in the provision of municipal services and the perceived quality in the citizen experience in the environmental management of the territory on the value relationship perceived by the citizen and general satisfaction. A second objective of the study is to see if there are significant differences in the hypotheses raised by gender by performing a multigroup analysis. This difference has been appreciated in two of the hypotheses. The study shows that the policies exercised by the territorial managers of the different areas have a significant influence on the value perceived by citizens, satisfaction and loyalty, which shape their general well-being. Areas for improvement in territorial policies and municipal services such as citizen security, air quality, public lighting and sports services have been identified. Knowing these shortcomings allows politicians to focus their efforts on improving the quality of life in cities.

## Introduction

Territorial and environmental policies as well as public services in urban areas have become a decisive factor when it comes to attracting and arranging the population. Proper management

**Funding:** The author(s) received no specific funding for this work.

**Competing interests:** The authors have declared that no competing interests exist.

of these policies is essential to achieve citizen satisfaction [1–3]. It is important to improve the expectations of the people who visit a city since it can be decisive in making the decision of establishing residence in the place [3–7]. The loyalty of citizens when establishing their residence in one city or another depends on the adequate provision of public services [5–9].

The correct planning of urban territorial policies, environment and municipal services promotes tourism, international sports activities and economic development [10, 11]. The expectations of citizens when establishing their residence is determined by the expectations of the standards of quality of life, the social welfare derived from public services, employment and entrepreneurship support, the preservation of cultural and heritage values and the environment [12, 13]. The construction of modern infrastructures, well dimensioned to the city and that solve the environmental impact problems, for which the territorial growth and infrastructures of urban areas must be balanced and compensated with environmental policies that reduce the pollution, noise and traffic that these infrastructures entail [14–16].

The citizen's environmental experience is another important factor when deciding whether to establish residence in and recommend the city. Public institutions manage green spaces and the rules that regulate maximum noise levels (noise pollution), allocate funds for cleaning and green points for the selective collection of urban waste. These define what quality green space is [17, 18]. There are several definitions of quality in the literature. A set of characteristics, which when combined, shows the appeal of green areas [19]. Show the attractiveness of green areas through various attributes that include size, variety of landscapes, cultural and historical aspects, tranquility and peace, as well as the quality of the facilities [20]. Studied the dimensions of perceived quality and activities in green spaces, using a composite park attractiveness index (which describes quality green spaces), which incorporates quality, three amenity factors, and two safety factors as indicators.

For a citizen to want to establish his residence in a metropolis, he considers the quality of life in it. The literature indicates that the larger the cities, the lower the level of satisfaction, due to insecurity and stress. Despite this, most people live in cities, due to the public services they offer in terms of quality and quantity. Rural areas are neglected in terms of basic services, their citizens having to move to the cities to satisfy medical, leisure, specialized supplies, etc [21].

The literature indicates that in rich countries rural life is preferred, while in poor or developing countries life in urban areas is preferred [22] In general, citizens think about money, creativity, municipal and health services, they attract them to the city without taking into account negative aspects such as crime, stress, traffic, pollution, the lack of green spaces [23].

The quality of life in urban areas is increasingly linked to political decisions about public services, the performance of municipal services has been one of the main concerns of public administrators and, therefore, has attracted considerable attention by the municipal administration. [24–26]. In recent decades, there has been a growing effort by social scientists to assess the quality of good governance in the city and the services it provides. Various research institutes, such as the Urban Institute and the International City Management Association (1974) [27] They have pioneered improvements in the provision of public services by collaborating with metropolitan governments to improve municipal service delivery systems. Academics like [28, 29] They have developed tools for validated measurements of the utility of the service, improving knowledge about the functioning of municipal services in urban areas. Other studies have attempted to evaluate various public services in terms of public satisfaction [30, 31].

Within the public services we have to include the quality of public transport, urban security, public lighting, water quality, sewerage and sewerage, funeral services, fire fighting service, etc.; services that together make up the perceived quality of a reception comfortable and safe city where to establish residence [9, 10, 32, 33].

Our analysis is supported by an index used in satisfaction surveys around the world. The American Customer Satisfaction Index [34, 35].

Our research brings a new perspective to the previous literature in relation to loyalty and citizen satisfaction in urban areas, since the expected quality is separated in territorial planning, municipal services and citizen's environmental experience [36, 37]. Once the citizens have decided to establish their residence in the urban area, the perceived quality is studied in the same dimensions. The internalized difference between perceived quality and expected quality leads citizens to shape the overall perceived value of the city in which they reside [38, 39]. The concept of moderator latent variable is also added to the classical theory of the Structural Equation Model [40, 41]. Three causal relationships study the modulating effect of the latent variables of perceived quality in territorial planning, municipal services and the citizen's environmental experience on the relationship that exists between the perceived value of the city and satisfaction measured as citizen well-being. [42, 43].

There are various reasons why citizens decide to live in metropolises; despite sacrificing uses related to happiness and quality of life, such as air quality and environmental quality, to achieve other advantages such as professional satisfaction, culture, money and social mobility [21].

The literature provides various studies that analyze these causal relationships, although it is true that few investigations are found that include modulating causal relationships between latent variables. Table 1 includes reference authors who use the items proposed for each of the latent variables [44, 45].

The latent variables that were studied are citizen satisfaction and loyalty [46, 47]. These are positively related and shows us the probability that a citizen has a high level of well-being measured as satisfaction in the set of municipal services and recommends the city as a place to live, attracting population to it, facing the problem of depopulation. from urban areas [48–69] and these affect the likelihood that citizens will stay and recommend the urban area, thus avoiding the dreaded depopulation [49].

The managers responsible for territorial policies in urban areas must have relevant information on the opinion of their citizens in order to plan actions that improve the expectations of potential new citizens attracted by them [9, 50, 51]. Those responsible for territorial policies in urban areas must have relevant information on the opinion of their citizens. These modify the behavior of citizens based on their municipal policies and services. [52, 53].

The city of Guayaquil is an appropriate territory to carry out this research: it is a port city, which opens doors to the beaches of the Pacific and the Galapagos Islands; it is the largest city and the city with the highest population density in the Republic of Ecuador, with 2'698,077 inhabitants, which corresponds to 15.6% of the Ecuadorian population. It is considered as the economic capital of the country for the dynamism generated in commercial, industrial, services and agricultural activity, it also has the largest municipal infrastructure in the country, responsible for providing public services to the citizens through 27 administrative and operational areas and public companies that do the work of providing services under municipal jurisdiction.

It is geographically divided into 16 urban and 5 rural areas. This research was carried out in the Tarqui urban parish, which houses 1'050,826 inhabitants, or 38.9% of the total population of Guayaquil and which is characterized by being a pole of urban development and greater economic growth, leading to a greater demand for quality public services.

This research studies the satisfaction of urban citizens measured as well-being derived from the decisions made by territorial and municipal managers. It also analyzes the loyalty of citizens residing in urban areas, measured as the recommendation of the quality of public services managed by the administrations and their managers, as well as the recommendation of

**Table 1. Scales used.**

| Reference | Dimension | Indicators |
|---|---|---|
| [57, 58] | (EQTPR) | (EQTPR1) Expectations Zoning and urban planning, (EQTPR2) Expectations Roads and pavements, (EQTPR3) Expectations Traffic organization and crossing, (EQTPR4) Expectations Public Transport service, (EQTPR5) Expectations Parking services, (EQTPR6) Expectations Address information, (EQTPR7) Expectations Transport terminal services |
| [10, 25, 31, 35, 36, 59, 60] | (EQMS) | (EQMS1) Expectations Drinking water quality, (EQMS2) Expectations Wastewater and sewerage services, (EQMS3) Expectations Garbage collection and environmental cleaning service, (EQMS4) Expectations Parks and gardens, (EQMS5) Expectations Street and road lighting, (EQMS6) Expectations Preservation of historical and cultural structures, (EQMS7) Expectations Cultural activities, (EQMS10) Expectations Social activities, (EQMS11) Expectations Firefighting services, (EQMS12) Expectations Municipal Police Services, (EQMS14) Expectations Veterinary Services. |
| [13, 14, 16, 61, 62] | (EQCE) | (EQCE1) Expectations Noise minimization, (EQCE2) Expectations Air pollution, (EQCE3) Expectations Green areas, (EQCE4) Expectations Recycling points. |
| [63, 64] | (PQTPR) | (PQTPR1) Perception Zoning and urban planning, (PQTPR2) Perception Roads and pavements, (PQTPR3) Perception Organization of traffic and crossing, (PQTPR4) Perception Public Transport Service, (PQTPR5) Perception Parking services, (PQTPR6) Perception Information of directions, (PQTPR7) Perception Transport terminal services |
| [65, 66] | (PQMS) | (PQMS1) Perception Quality of drinking water, (PQMS2) Perception Wastewater and sewerage services, (PQMS3) Perception Garbage collection and environmental cleaning service, (PQMS4) Perception Parks and gardens, (PQMS5) Perception Public and road lighting, (PQMS6) Perception Preservation of historical and cultural structures, (PQMS7) Perception Cultural activities, (PQMS8) Perception Social and cultural facilities, (PQMS9) Perception Cemetery services, (PQMS10) Perception Social activities, (PQMS11) Perception Firefighting services, (PQMS12) Perception Municipal police services, (PQMS13) Perception Sports services, (PQMS14) Perception Veterinary services |
| [67, 68] | (PQCE) | (PQEC1) Perception Noise minimization, (PQEC2) Perception Air pollution, (PQEC3) Perception Green areas, (PQEC4) Perception recycling points |
| [39, 69] | (PV) | (PV1) Perceived value Public Transport Service, (PV2) Perceived value Parking services, (PV3) Perceived value Transport terminal services, (PV4) Perceived value Drinking water quality, (PV5) Perceived value Wastewater and sewerage services, (PV6) Perceived value Garbage collection and environmental cleaning service, (PV7) Perceived value Parks and gardens, (PV8) Perceived value Public and road lighting, (PV9) Perceived value Preservation of historical and cultural structures, (PV10) Perceived value Cultural activities, (PV11) Perceived value Social and cultural facilities, PV12) Perceived value Cemetery services(PV13) Perceived value Social activities,(PV14) Perceived value Firefighting services, (PV15) Perceived value Municipal police services, (PV16) Perceived value Sports services, (PV17) Perceived value Veterinary services |
| [70, 71] | (SATISFAC) | (S1) Satisfaction in the experience of territorial planning and road, (S2) Satisfaction in the experience of the provision of municipal services, (S3) Overall satisfaction in the user experience |
| [7, 29, 32, 72–74] | (LOYALTY) | (L1) Loyalty I would recommend to my friends or family to vote for the mayor of Guayaquil (L2) Loyalty I would vote again for the mayor of Guayaquil (L3) Loyalty I would recommend the services provided by the Municipality of Guayaquil (L4) Loyalty I would recommend to family or friends living outside the city to move to live in Guayaquil (L5) Loyalty I would recommend to family or friends who live outside the city to do tourism in Guayaquil |

residents to other citizens, whether they are residents of urban areas or residents of other urban areas. In this way satisfied and loyal citizens contribute to the fight against the depopulation of vast urban areas, a problem that greatly worries public administrations. This study is also innovative because it uses a model of structured equations with moderating variables.

That is, given the different values taken by the variables of perceived quality (PQTPR, PQMS, PQEEC), how the direct relationship moderate between the perceived value of citizens in municipal services and citizen satisfaction measured as their well-being.

## Materials and methods

The variables used to measure the fidelity of the citizens residing in Guayaquil, 1) expected quality territorial and road planning (EQTPR) 2) the expected quality provision of municipal services (EQMS), 3) expected quality experience of the environmental citizen (EQEEC), 4) perceived quality territorial planning and road (PQTPR), 5) perceived quality provision of municipal services (PQMS), 6) perceived quality experience of the environmental citizen (PQEEC), 7) the perceived value of the citizen (PV), 8) satisfaction (SATISFAC) and 9) loyalty (LOYALTY).

The theoretical hypotheses (Fig 1) are raised in accordance with the theoretical framework described:

Hypothesis 1 (H1). The expectation of citizens in urban areas regarding the experience in territorial and road planning (EQTPR) influences directly and significantly the perceived quality of the city's territorial and road planning experience (PQTPR).

Hypothesis 2 (H2). The expectation of the experience of providing municipal services in urban areas (EQMS) influences directly and significantly the perceived quality of the experience of providing municipal services in the city (PQMS).

Hypothesis 3 (H3). The expectations of the environmental experience of the citizen of urban areas (EQCE) influences directly and significantly the perceived quality of the environmental experience of the citizen in the city (PQCE).

Hypothesis 4 (H4). The perceived quality of territorial and road planning in urban areas (PQTPR) influences directly and significantly the overall perceived value of the city (PV).

Hypothesis 5 (H5). The quality of municipal service provision in urban areas (PQMS) influences directly and significantly the overall perceived value of the city (PV).

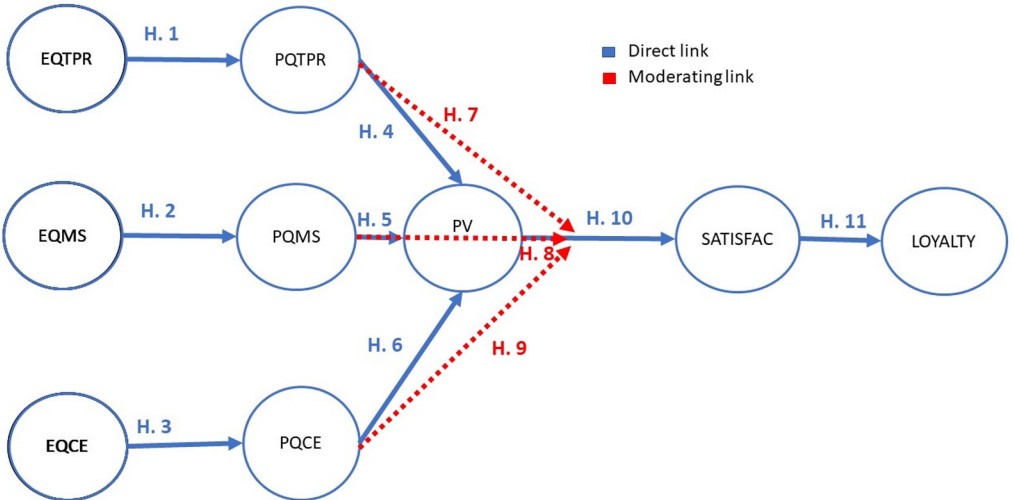

**Fig 1. Theoretical model.**

Hypothesis 6 (H6). The perceived quality of the environmental experience of citizens in urban areas (PQCE) influences directly and significantly the general perceived value of the city (PV).

Hypothesis 7 (H7). The perceived quality of the territorial planning experience in urban areas (PQTPR) directly and significantly modulates the direct relationship between the perceived value of the urban municipality (PV) and citizen satisfaction (SATISFAC).

Hypothesis 8 (H8). The perceived quality of the experience in the provision of municipal services in urban areas (PQMS) directly and significantly modulates the direct relationship between the perceived value of the city (PV) and citizen satisfaction (SATISFAC).

Hypothesis 9 (H9). The perceived quality of the citizen's environmental experience in urban areas (PQCE) directly and significantly modulates the direct relationship between the perceived value of the citizen (PV) and the general satisfaction of the citizen in the city. (SATISFAC).

Hypothesis 10 (H10). The global value perceived by the citizen in urban areas (PV) influences directly and significantly the general satisfaction of the citizen. (SATISFAC).

Hypothesis 11 (H11). General citizen satisfaction in the urban municipal service (SATISFAC) influences directly and significantly the general loyalty of the citizen (LOYALTY).

## Data

This research is aimed at establishing the causal relationships between latent variables based on a study of the expected quality with respect to the perceived quality, perceived value, level of satisfaction and loyalty of municipal public services in the urban sector Tarqui, characterized by being the largest in terms of territorial extension, as well as population density and economic growth of the city of Guayaquil.

The data were obtained from an opinion, containing 80 questions, classified into 9 constructs, conducted at the household level. The universe of study is formed by 1′050.826 inhabitants that corresponds to the 38.9% of the total population of Guayaquil, and considering limitations of time and resources, we calculated a simple random sample representative of 528 households with a margin of error of 5% and level of confidence of 95%, achieving favorable responses of 521 families, surveying a representative from each household.

The data were collected by a group of qualified interviewers from October 5 to 11, 2020, obtaining relevant sociodemographic information, as well as the levels of quality expected (expectation), perceived, satisfaction and loyalty of the respondents in each item contained in the questionnaire, implementing a Likert scale ranging from one to seven for each proposed statement [54].

The sample covers the entire spectrum of age, level of education, professional category and income level [55]. The validity of the questionnaire was ensured using questionnaires of reference authors in the field of study of satisfaction [56].

## Methodology

The questionnaire consists of five sections:

Section 1: Experience in territorial planning and road planning. Questions about Zoning and urban planning, Roads and pavements, Traffic organization and crossing, Public Transport Service, Parking services, Address information, Transport terminal services, Global satisfaction in the experience of territorial planning and road.

Section 2: Experience in the provision of municipal services. Questions about Quality/Price Ratio, Drinking Water Quality, Wastewater and Sewerage Services, Garbage Collection and Environmental Cleaning Service, Parks and Gardens, Street and Road Lighting, Preservation of Historical and Cultural Structures, Cultural Activities, Social and Cultural Facilities, Cemetery Services, Social Activities, Firefighting Services, Municipal Police Services, Sports Services, Veterinary Services, Global Satisfaction in the experience of the municipal service provision.

Section 3: Environmental experience of the citizen. Questions about Noise Minimization, Air Pollution, Green Areas, Recycling Points, Global Satisfaction in the User's Environmental Experience.

Section 4: Loyalty of the citizen. Questions about whether they would recommend the services provided by the Municipality of Guayaquil, if they would recommend to family or friends who live outside the city to move to live in Guayaquil, if they would recommend to family or friends who live outside the city to do tourism in Guayaquil, if they would recommend my friends or family to vote for the mayor of Guayaquil, if they would vote again for the mayor of Guayaquil.

Section 5: Sociodemographic. Profile, Gender, age, level of education, professional category, level of family income.

At all times, respondents were informed about the academic purpose of the study, in the same way that their responses would be anonymous. Before starting the questions, they were asked permission for their answers to be part of the study. To carry out the research, questions with a Likert scale were set. The survey haved 80 items, the sample data was collected at different times of the day. The survey was given to residents of Guayaquil. The sociodemographic profile was completed with closed questions. 521 valid questionnaires were obtained in the sample, which had a confidence level of 95% and a sampling error of 3.25%. The Warl PLS program has been used for the validation of the model. Table 1 shows the reference authors, the latent variables and the items that make up each of these variables.

## Results

Once the methodology of the proposed structural equation model has been applied, results have been obtained that will be validated one by one.

From the survey data, the sociodemographic profile has been calculated (Table 2). 46.9% of the interviewees were women, compared to 54.1% of the men.Most of those surveyed are young people under 30 years of age (44.7%) with secondary education (51.1%).

Table 3 shows the structural coefficients of the items of each construct.

### Individual reliability of the observed variables

The proposed model was validated by studying the validity and reliability of the latent variables.

The factor loadings and the limiting probability of the observed variables are shown in Table 4. Once the individual reliability of the items had been analyzed, the validity and reliability of the latent variables of the model was studied.

In order to study coliniality, the value of the variance of the inflation factor (VIF) was verified, whose value must be greater than five (Table 5).

The analysis of all these values validates the variables used, the factorial loads have values equal to or greater than 0.505 according to [75].

**Table 2. Sociodemographic profile.**

| Variable | Category | Absolute frequency | Percentage |
|---|---|---|---|
| **Gendern = 521)** | Male | 282 | 54.1 |
| | Female | 239 | 45.9 |
| **Age (n = 521)** | | | |
| | >30 | 233 | 44.7 |
| | [30–39] | 165 | 31.7 |
| | [40–49] | 67 | 12.9 |
| | [50–59] | 40 | 7.7 |
| | 60 or more | 16 | 3.1 |
| **Studies (n = 521)** | | | |
| | Without Studies | 1 | 0.2 |
| | Primary School | 36 | 6.9 |
| | Secondary school | 266 | 51.1 |
| | University | 207 | 39.7 |
| | Postgraduate | 11 | 2.2 |
| **Family Income (n = 521)** | | | |
| | Less than 400 dollars | 367 | 70.4 |
| | Between 400 and 2000 dollars | 146 | 28 |
| | Between 2001 and 5000 dollars | 4 | 0.8 |
| | Between 5001 and 10000 dollars | 2 | 0.4 |
| | More than 10001 dollars | 2 | 0.4 |

The study of all these items does not show statistically significant loads at 99.99%, therefore we have considered the model valid and reliable.

## Analysis of composite reliability of constructs

In order to measure whether the observed items strictly and appropriately measure the latent variable of which they are part, the composite reliability values as well as Cronbach's Alpha are studied, verifying if they are greater than / equal to 0.7 (Table 6).

## Convergent validity

The convergent validity of a set of items is measured with the mean variance extracted (AVE) (Table 7). Fornell and Larcker (1981) [104] determined that the minimum value of the AVE must be greater than 0.5, which means that the indicators are shared more than half of the variance of constructs [75].

## Hypothesis testing

Table 8 shows the various contrasts of the parameters used in order to verify the adequate fit of the proposed model.

After having analyzed all the latent variables, considering their validity, as well as an adequate adjustment, it can be confirmed that the results obtained are reliable and justify their applicability.

To finalize the hypothesis test, the factor loadings of each causal relationship were calculated (Table 9).

The hypotheses (H1), (H2), (H3), (H4), (H5), (H6), (H7), (H9), (H10) and (H11) were confirmed by the contract carried out. The factorial loads of the causal relationships, both direct

**Table 3. Structural coefficients of the items and limit probability.**

| Latent variable | Observed variable | Standardized coefficient | p-Value | Latent variable | Observed variable | Standardized coefficient | p-Value |
|---|---|---|---|---|---|---|---|
| EQTPR | EQTPR1 | 0.414 | <0.001 | PQMS | PQMS1 | 0.420 | <0.001 |
| | EQTPR2 | 0.424 | <0.001 | | PQMS2 | 0.444 | <0.001 |
| | EQTPR3 | 0.438 | <0.001 | | PQMS3 | 0.394 | <0.001 |
| | EQTPR4 | 0.415 | <0.001 | | PQMS4 | 0.420 | <0.001 |
| | EQTPR5 | 0.445 | <0.001 | | PQMS5 | 0.426 | <0.001 |
| | EQTPR6 | 0.435 | <0.001 | | PQMS6 | 0.433 | <0.001 |
| | EQTPR7 | 0.398 | <0.001 | | PQMS7 | 0.409 | <0.001 |
| EQMS | EQMS1 | 0.400 | <0.001 | | PQMS8 | 0.419 | <0.001 |
| | EQMS2 | 0.412 | <0.001 | | PQMS9 | 0.472 | <0.001 |
| | EQMS3 | 0.403 | <0.001 | | PQMS10 | 0.427 | <0.001 |
| | EQMS4 | 0.409 | <0.001 | | PQMS11 | 0.462 | <0.001 |
| | EQMS5 | 0.403 | <0.001 | | PQMS12 | 0.478 | <0.001 |
| | EQMS6 | 0.409 | <0.001 | | PQMS13 | 0.402 | <0.001 |
| | EQMS7 | 0.412 | <0.001 | | PQMS14 | 0.374 | <0.001 |
| | EQMS10 | 0.416 | <0.001 | PQEC | PQEC15 | 0.458 | <0.001 |
| | EQMS11 | 0.420 | <0.001 | | PQEC16 | 0.587 | <0.001 |
| | EQMS12 | 0.431 | <0.001 | | PQEC17 | 0.469 | <0.001 |
| | EQMS13 | 0.413 | <0.001 | | PQEC18 | 0.517 | <0.001 |
| | EQMS14 | 0.416 | <0.001 | PV | PV1 | 0.363 | <0.001 |
| EQCE | EQCE1 | 0.449 | <0.001 | | PV2 | 0.371 | <0.001 |
| | EQCE2 | 0.445 | <0.001 | | PV3 | 0.381 | <0.001 |
| | EQCE3 | 0.432 | <0.001 | | PV4 | 0.434 | <0.001 |
| | EQCE4 | 0.441 | <0.001 | | PV5 | 0.399 | <0.001 |
| PQTPR | PQTPR1 | 0.454 | <0.001 | | PV6 | 0.415 | <0.001 |
| | PQTPR2 | 0.479 | <0.001 | | PV6 | 0.393 | <0.001 |
| | PQTPR3 | 0.509 | <0.001 | | PV8 | 0.424 | <0.001 |
| | PQTPR4 | 0.454 | <0.001 | | PV9 | 0.429 | <0.001 |
| | PQTPR5 | 0.413 | <0.001 | | PV10 | 0.434 | <0.001 |
| | PQTPR6 | 0.396 | <0.001 | | PV11 | 0.490 | <0.001 |
| | PQTPR7 | 0.408 | <0.001 | | PV12 | 0.438 | <0.001 |
| LOYALTY | L1 | 0.616 | <0.001 | | PV13 | 0.419 | <0.001 |
| | L2 | 0.644 | <0.001 | | PV14 | 0.417 | <0.001 |
| | L3 | 0.619 | <0.001 | | PV15 | 0.443 | <0.001 |
| | L4 | 0.580 | <0.001 | | PV16 | 0.402 | <0.001 |
| | L5 | 0.519 | <0.001 | | PV17 | 0.399 | <0.001 |
| | L4 | 0.580 | <0.001 | SATISFAC | S1 | 0.500 | <0.001 |
| | | | | | S2 | 0.533 | <0.001 |
| | | | | | S3 | 0.537 | <0.001 |

and modulating, are shown in Fig 2. The p-value of each one allows us to ascertain whether a hypothesis is supported or not.

## Discussion and conclusion

The satisfaction of citizens residing in urban areas is a determining factor for loyalty and is rooted in their territory. These urban citizens recommend and share their experience in this type of environment with family, friends and co-workers. In this way, the depopulation of urban areas can be fought with actions aimed at increasing the satisfaction of resident citizens

**Table 4. Indicator loading of the observed variables.**

| Latent Variable | Observed variables | Indicator loading | *p*-Value |
|---|---|---|---|
| EQTPR | (EQTPR1) Expectations Zoning and urban planning | 0.864 | <0.001 |
| | (EQTPR2) Road and Pavement Expectations | 0.904 | <0.001 |
| | (EQTPR3) Expectations Traffic organization and crossing | 0.911 | <0.001 |
| | (EQTPR4) Expectations Public Transport Service | 0.901 | <0.001 |
| | (EQTPR5) Expectations Parking Services | 0.884 | <0.001 |
| | (EQTPR6) Expectations Address Information | 0.895 | <0.001 |
| | (EQTPR7) Expectations Transport terminal services | 0.903 | <0.001 |
| EQMS | (EQMS1) Expectations Drinking water quality | 0.881 | <0.001 |
| | (EQMS2) Expectations Wastewater and Sewerage Services | 0.884 | <0.001 |
| | (EQMS3) Expectations Garbage collection service and environmental cleaning | 0.865 | <0.001 |
| | (EQMS4) Expectations Parks and gardens | 0.865 | <0.001 |
| | (EQMS5) Expectations Street and road lighting | 0.902 | <0.001 |
| | (EQMS6) Expectations Preservation of historical and cultural structures | 0.879 | <0.001 |
| | (EQMS10) Expectations Cemetery Services | 0.899 | <0.001 |
| | (EQMS11) Expectations Social activities | 0.886 | <0.001 |
| | (EQMS12) Expectations Firefighting Services | 0.838 | <0.001 |
| | (EQMS14) Expectations Sports services | 0.879 | <0.001 |
| EQCE | (EQCE1) Expectations Noise minimization | 0.897 | <0.001 |
| | (EQCE2) Expectations Air pollution | 0.932 | <0.001 |
| | (EQCE3) Expectations Green Areas | 0.920 | <0.001 |
| | (EQCE4) Expectations Recycling points | 0.915 | <0.001 |
| PQTPR | (PQTPR1) Perception Zoning and urban planning | 0.805 | <0.001 |
| | (PQTPR2) Perception Roads and pavements | 0.770 | <0.001 |
| | (PQTPR3) Perception Organization of traffic and crossing | 0.734 | <0.001 |
| | (PQTPR4) Perception Public Transport Service | 0.780 | <0.001 |
| | (PQTPR5) Percepción Parking services | 0.733 | <0.001 |
| | (PQTPR6) Perception Information of direction | 0.758 | <0.001 |
| | (PQTPR7) Perception Transport terminal services | 0.808 | <0.001 |
| PQMS | (PQMS1) Perception of drinking water quality | 0.655 | <0.001 |
| | (PQMS2) Perception Wastewater and sewerage services | 0.650 | <0.001 |
| | (PQMS3) Perception Garbage collection and environmental cleaning service | 0.739 | <0.001 |
| | (PQMS4) Perception Parks and gardens | 0.707 | <0.001 |
| | (PQMS5) Perception of public and road lighting | 0.679 | <0.001 |
| | (PQMS6) Perception Preservation of historical and cultural structures | 0.766 | <0.001 |
| | (PQMS7) Perception Cultural activities | 0.775 | <0.001 |
| | (PQMS8) Perception Social and cultural facilities | 0.799 | <0.001 |
| | (PQMS9) Perception Cemetery Services | 0.690 | <0.001 |
| | (PQMS10) Perception Social activities | 0.758 | <0.001 |
| | (PQMS11) Perception of firefighting services | 0.767 | <0.001 |
| | (PQMS12) Perception Municipal police services | 0.573 | <0.001 |
| | (PQMS13) Perception Sports services | 0.745 | <0.001 |
| | (PQMS14) Perception Veterinary services | 0.651 | <0.001 |
| PQEC | (PQEC1) Perception Noise minimization | 0.778 | <0.001 |
| | (PQEC2) Perception Air pollution | 0.800 | <0.001 |
| | (PQEC3) Perception Green areas | 0.823 | <0.001 |
| | (PQEC4) Perception recycling points | 0.798 | <0.001 |

(*Continued*)

**Table 4.** (Continued)

| Latent Variable | Observed variables | Indicator loading | p-Value |
|---|---|---|---|
| PV | (PV1) Perceived value Public Transport Service | 0.759 | <0.001 |
| | (PV2) Perceived value Parking services | 0.747 | <0.001 |
| | (PV3) Perceived value Transport terminal services | 0.634 | <0.001 |
| | (PV4) Perceived value Drinking water quality | 0.758 | <0.001 |
| | (PV5) Perceived value Wastewater and sewerage services | 0.797 | <0.001 |
| | (PV6) Perceived value Garbage collection service and environmental cleaning | 0.788 | <0.001 |
| | (PV7) Perceived value Parks and gardens | 0.606 | <0.001 |
| | (PV8) Perceived value Public and road lighting | 0.775 | <0.001 |
| | (PV9) Perceived value Preservation of historical and cultural structures | 0.695 | <0.001 |
| | (PV10) Perceived value Cultural activities | 0.730 | <0.001 |
| | (PV11) Perceived value Social and cultural facilities | 0.557 | <0.001 |
| | (PV12) Perceived value Cemetery services | 0.629 | <0.001 |
| | (PV13) Perceived value Social activities | 0.646 | <0.001 |
| | (PV14) Perceived value Fire extinguishing services | 0.641 | <0.001 |
| | (PV15) Perceived value Municipal police services | 0.762 | <0.001 |
| | (PV16) Perceived value Sports services | 0.529 | <0.001 |
| | (PV17) Perceived value Veterinary services | 0.634 | <0.001 |
| SATISFAC | (S1) Satisfaction in the experience of territorial planning and road | 0.842 | <0.001 |
| | (S1) Satisfaction in the experience of the provision of municipal services | 0.901 | <0.001 |
| | (S3) Overall user experience satisfaction | 0.891 | <0.001 |
| LOYALTY | (L1) Loyalty I would recommend to my friends or family to vote for the mayor of Guayaquil | 0.907 | <0.001 |
| | (L2) Loyalty I would vote again for the mayor of Guayaquil | 0.887 | <0.001 |
| | (L3) Loyalty I would recommend the services provided by the Municipality of Guayaquil | 0.905 | <0.001 |
| | (L4) Loyalty I would recommend to family or friends living outside the city to move to live in Guayaquil | 0.834 | <0.001 |
| | (L5) Loyalty I would recommend to family or friends who live outside the city to do tourism in Guayaquil | 0.806 | <0.001 |

in the various areas that the territory managers serve, in territorial planning and viability, in the experience as a user of municipal services and in the field of environmental policies carried out. The public administration has, among others, the objective of preserving urban areas, maintaining the sustainable value of these residential destinations that it manages and establishing and becoming attractive for new residents. In this way, the resources available to improve municipal services and policies to improve citizen satisfaction would be increased [9].

The analysis of the fidelity of citizens in urban areas is important for the design and strategic planning of the territory. In this analysis, a model of structural equations was implemented in which modulating causal relationships were included on the relationship between satisfaction and perceived value. The Fidelity of the resident citizens in the urban residential destination of Guayaquil was analyzed. The results obtained in our analysis confirmed most of the causal relationships in the proposed model and can be used to improve the quality of life of citizens in urban areas. When citizens choose an urban population as a residential destination, once installed, they evaluate the gap between the expected quality of all areas of their population (territorial planning and roads, municipal services, and the environment) and their perceived quality. To choose an area for residence, before setting up residence in a city, the citizen is informed about the quality of life in it, the services they offer and with all this information generates an idea of what life is like in the residential destination. If the perceived quality in your urban city is equal to or higher than the quality expected before choosing this destination to establish your residence, the perceived value in the set of municipal public services, and

**Table 5. Individual VIF of the indicators.**

| Latent Variable | Observaded Varible | VIF | Latent Variable | Observaded Varible | VIF |
|---|---|---|---|---|---|
| EQTPR | EQTPR1 | 3.101 | PQMS | PQMS1 | 0.420 |
| | EQTPR2 | 4.124 | | PQMS2 | 0.444 |
| | EQTPR3 | 4.389 | | PQMS3 | 0.394 |
| | EQTPR4 | 4.195 | | PQMS4 | 0.420 |
| | EQTPR5 | 3.569 | | PQMS5 | 0.426 |
| | EQTPR6 | 3.868 | | PQMS6 | 0.433 |
| | EQTPR7 | 4.248 | | PQMS7 | 0.409 |
| EQMS | EQMS1 | 4.488 | | PQMS8 | 0.419 |
| | EQMS2 | 4.820 | | PQMS9 | 0.472 |
| | EQMS3 | 4.069 | | PQMS10 | 0.427 |
| | EQMS4 | 3.623 | | PQMS11 | 0.462 |
| | EQMS5 | 4.560 | | PQMS12 | 0.478 |
| | EQMS6 | 4.217 | | PQMS13 | 0.402 |
| | EQMS7 | 3.882 | | PQMS14 | 0.374 |
| | EQMS10 | 0.416 | PQEC | PQEC15 | 0.458 |
| | EQMS11 | 0.420 | | PQEC16 | 0.587 |
| | EQMS12 | 0.431 | | PQEC17 | 0.469 |
| | EQMS13 | 0.413 | | PQEC18 | 0.517 |
| EQCE | EQCE1 | 0.449 | PV | PV1 | 0.363 |
| | EQCE2 | 0.445 | | PV2 | 0.371 |
| | EQCE3 | 0.432 | | PV3 | 0.381 |
| | EQCE4 | 0.441 | | PV4 | 0.434 |
| PQTPR | PQTPR1 | 0.454 | | PV5 | 0.399 |
| | PQTPR2 | 0.479 | | PV6 | 0.415 |
| | PQTPR3 | 0.509 | | PV6 | 0.393 |
| | PQTPR4 | 0.454 | | PV8 | 0.424 |
| | PQTPR5 | 0.413 | | PV9 | 0.429 |
| | PQTPR6 | 0.396 | | PV10 | 0.434 |
| | PQTPR7 | 0.408 | | PV11 | 0.490 |
| LOYALTY | L1 | 0.616 | | PV12 | 0.438 |
| | L2 | 0.644 | | PV13 | 0.419 |
| | L3 | 0.619 | | PV14 | 0.417 |
| | L4 | 0.580 | | PV15 | 0.443 |
| | L5 | 0.519 | | PV16 | 0.402 |
| SATISFAC | S1 | 0.500 | | PV17 | 0.399 |
| | S2 | 0.533 | SATISFAC | S3 | 0.537 |

your experience as a user of territorial and environmental policies is high. This set of experiences, experiences and perceptions lead us to satisfaction measured as the well-being of the urban citizen, which leads us to recommend the urban area as a residential destination to other citizens [76, 77].

Citizens of urban areas have very high expectations of public services, territorial planning and viability, as well as environmental experience, obtaining the maximum position in all measured dimensions. Once citizens have decided to establish their residence and have experienced, enjoyed and used the various public services, they perceive their quality. The main reasons why citizens of urban areas are satisfied and recommend family and friends to establish their residence in the urban area have been parking services, firefighting service, quality of

**Table 6. Composite reliability and cronbach's alpha.**

|  | Composite reliability | Cronbach's alpha |
|---|---|---|
| PV | 0.939 | 0.930 |
| SATISFAC | 0.910 | 0.851 |
| LOYALTY | 0.939 | 0.918 |
| EQTRP | 0.966 | 0.958 |
| PQTRP | 0.911 | 0.911 |
| EQMS | 0.973 | 0.973 |
| PQMS | 0.935 | 0.935 |
| EQCE | 0.954 | 0.954 |
| PQCE | 0.877 | 0.877 |
| PQTRP*PV | 1.000 | 1.000 |
| PQCE*PV | 1.000 | 1.000 |
| PQMS*PV | 1.000 | 1.000 |

drinking water, services wastewater and sewage, city cleaning and garbage collection, parks and gardens, preservation of historical and cultural monuments, cultural activities, social activities, sports services and air quality [78–81].

Hypothesis 1. The quality expected in the territorial planning and the roads of the citizen residing in an urban area has a directly and significantly influence on their perceived quality of life. In Fig 3 the causal relationship between the two variables can be observed. For very high quality values expected by citizens, their perceived quality decreases. This result is in agreement with studies. The managers of planning and territorial viability should try to carry out policies in urban areas that aim at social welfare, avoid inequality and generate sustainable employment. The items in which they can be improved in the opinion of the Yu et al., (2018) and Afacan (2015) [82, 83].

Hypothesis 2. The expected quality of municipal services for citizens living in urban areas directly and significantly influences their perceived quality. Fig 4 shows that very high values of expected quality have a positive influence on perceived quality. This result is in the same line of studies by Municipal services administrators that meet the necessary quality as a whole so that residents do not decide to migrate to another residential destination. The items that stand out especially for their low score, and therefore clearly improvable by municipal and

**Table 7. Average variance extracted.**

| Average Variance Extracted (AVE) | |
|---|---|
| EQTRP | 0.800 |
| PQTRP | 0.593 |
| EQMS | 0.769 |
| PQMS | 0.509 |
| EQCE | 0.840 |
| PQCE | 0.640 |
| PV | 0.479 |
| SATISFAC | 0.771 |
| LOYALTY | 0.755 |
| PQTRP*PV | 1.000 |
| PQMS*PV | 1.000 |
| PQCE*PV | 1.000 |

**Table 8. Goodness-of-fit.**

| | | |
|---|---|---|
| Average path coefficient (APC) | 0.427 | P<0.001 |
| Average R-squared (ARS) | 0.559 | |
| Average adjusted R-squared (AARS) | 0.558 | |
| Average block VIF (AVIF) | 3.008 | acceptable if < = 5, ideally < = 3.3 |
| Average full collinearity VIF (AFVIF) | 4.807 | |
| Tenenhaus GoF (GoF) | 0.653 | small > = 0.1, medium > = 0.25, large > = 0.36 |
| Sympson's paradox ratio (SPR) | 0.909 | acceptable if > = 0.7, ideally = 1 |
| Standardized threshold difference count ratio (STDCR) | 0.984 | |
| Standardized threshold difference sum ratio (STDSR) | 0.922 | |
| R-squared contribution ratio (RSCR) | 0.992 | acceptable if > = 0.9, ideally = 1 |
| Statistical suppression ratio (SSR) | 1.000 | acceptable if > = 0.7 |
| Nonlinear bivariate causality direction ratio (NLBCDR) | 0.909 | |
| Standardized root mean squared residual (SRMR) | 0.081 | acceptable if < = 0.1 |
| Standardized mean absolute residual (SMAR) | 0.061 | |

state managers, are firstly the municipal police services, followed by air pollution and noise pollution [81, 84].

Hypothesis 3. The expected quality of the user experience of the environmental citizen residing in urban areas has a directly and significantly influence on its perceived quality. Fig 5 shows the flattened sinusoidal behavior to the right of this variable, and it is shown that for very high values of expected quality the trend is change. Environmental policy managers must guide them to turn urban cities into sustainable destinations, and benchmarks in recycling, noise reduction, clean points, green areas and pollution that meet the necessary quality as a whole so that residents do not decide to emigrate to another residential destination [85–87].

Hypothesis 4. The perceived quality of territorial planning and the viability of the citizen residing in an urban Area directly and significantly influences the The perceived quality of territorial planning of the destination as a whole [87–89]. Fig 6 shows the direct influence of quality in urban planning and viability on the total perceived value. The administrators of planning and territorial feasibility must

**Table 9. Hypothesis testing.**

| Hypothesis | Effect | Path Coefficient | p-Value | Supported? |
|---|---|---|---|---|
| H1: EQTRP-PQTRP | + | 0.752 | <0.001*** | YES |
| H2: EQMS-PQMS | + | 0.771 | <0.001*** | YES |
| H3: EQCE-PQCE | + | 0.614 | <0.001*** | YES |
| H4: PQTRP-PV | + | 0.128 | 0.002** | YES |
| H5: PQMS-PV | + | 0.755 | <0.001*** | YES |
| H6: PQCE-PV | + | 0.100 | 0.010** | YES |
| H7: PQTRP->PV-SATISFAC | + | 0.072 | 0.050* | YES |
| H8: PQMS-> PV- SATISFAC | + | -0.020 | 0.322 | NO |
| H9: PQCE-> PV-SATISFAC | + | 0.121 | 0.003** | YES |
| H10: PV-SATISFAC | + | 0.648 | <0.001*** | YES |
| H11: SATISFAC—LOYALTY | + | 0.719 | <0.001*** | YES |

a = 0,001 (***),

a = 0,01 (**),

a = 0,05 (*).

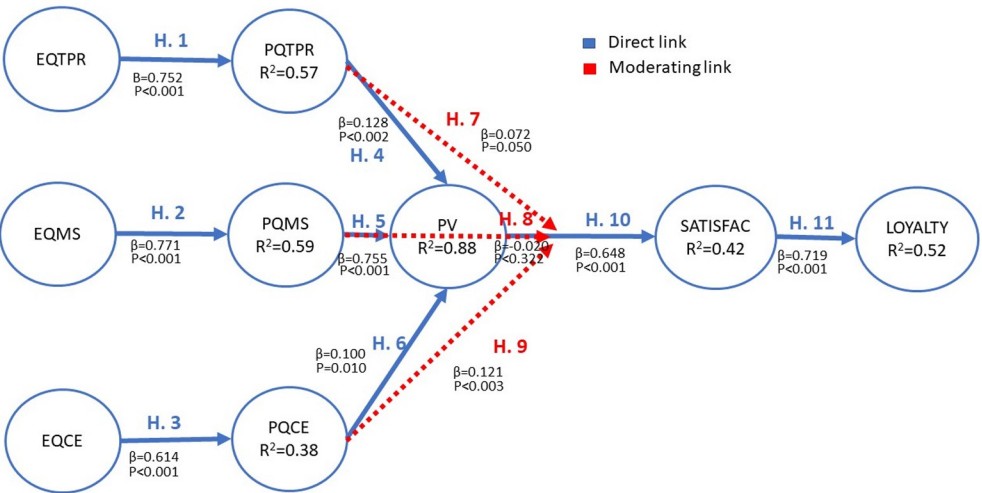

**Fig 2. Loading factor and limiting probability of the proposed causal relationships.**

be aware that investment in infrastructure for public use must be adequate and proportionate to the locality in which it is located, always promoting accessibility and comfort for the residents of the destination, causing a feeling of comfort and well-being that makes citizens feel proud and thus recommend the city [89, 90].

Hypothesis 5. Hypothesis 5. The perceived quality of municipal services by citizens living in urban areas has a directly and significantly influence on the perceived value of the destination as a whole [91–93]. Fig 7 shows the direct influence of perceived quality in municipal public services on the total perceived value. This result is in agreement with the research of Oriade, A and Schofield, P (2019) [94]. Municipal service managers must provide adequate public services to their population and estimate future demand so as not to collapse services, providing satisfaction, well-being and roots for citizens [95, 96].

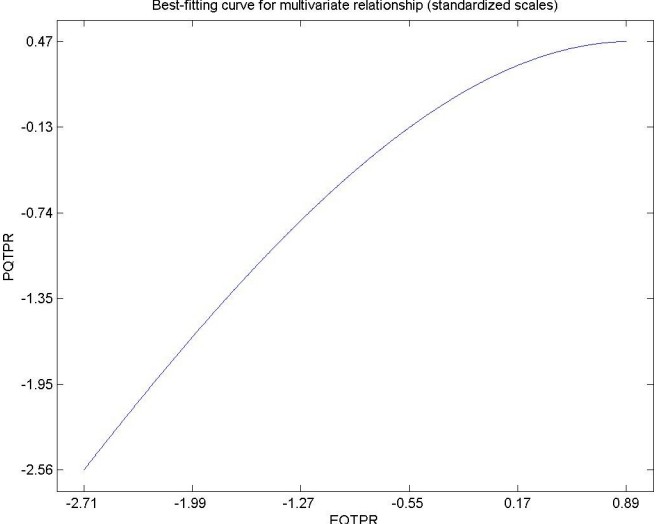

**Fig 3. Hypothesis 1, EQTRP-PQTRP.**

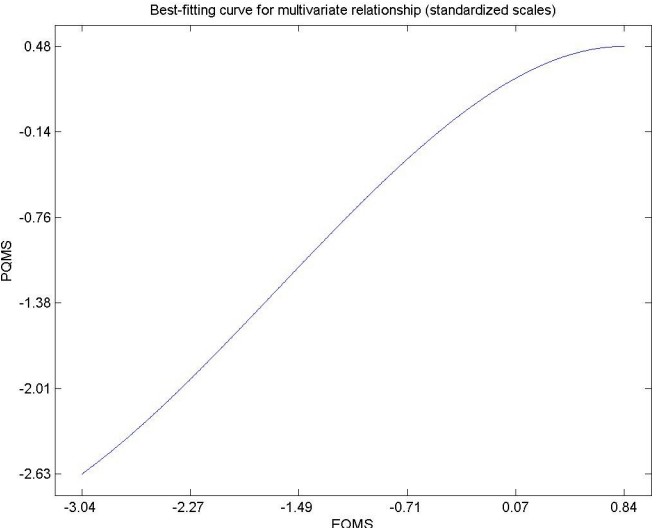

**Fig 4. Hypothesis 2, EQMS-PQMS.**

Hypothesis 6. The perceived quality of the environmental experience by citizens living in urban areas has a directly and significantly influence on the perceived value [97–99]. Fig 8 shows the positive influence of perceived quality on the perceived value, for very high values of perceived quality in municipal services the influence is decreasing. This result is in agreement with the research of [96]. Municipal service managers must provide adequate public services to their population and estimate future demand so as not to collapse services, providing satisfaction, well-being and roots for citizens [100, 101].

Hypothesis 7 (Fig 9A.) The perceived quality of land and road planning by citizens in urban areas directly and significantly modulates causal relation between perceived value and satisfaction measured as well-being in the resident's locality [102, 103]. Fig 9B shows a sinusoidal shape in which intensity is lost in the external areas, in the relationship between perceived

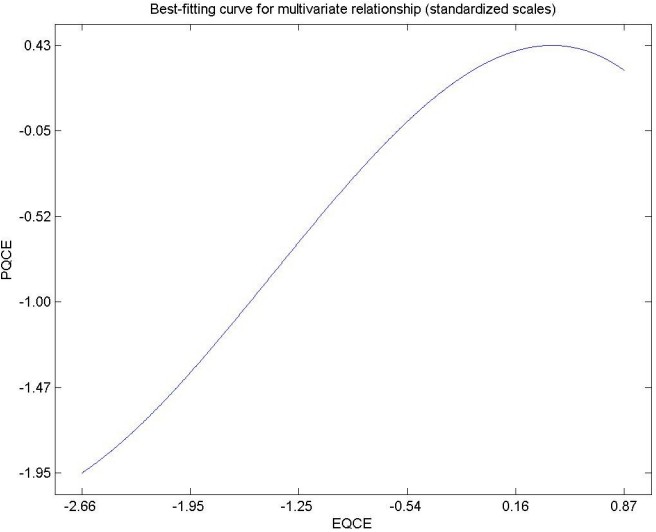

**Fig 5. Hypothesis 3, EQCE-PQCE.**

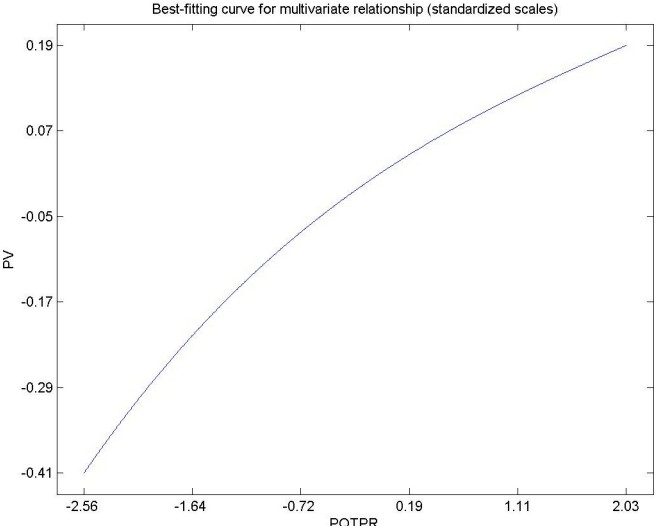

**Fig 6. Hypothesis 4, PQTRP-PV.**

value and quality, both for high values and when the low values of the moderating variable (perceived quality in territorial planning and viability). Confirmation of this hypothesis means that by increasing the quality of territorial planning and viability, not only the perceived value is influenced but also the relationship between the perceived value and satisfaction, thus increasing the quality of territorial planning. This helps increase overall satisfaction of the citizens residing in urban areas with the experience.

Hypothesis 8 (Fig 10A) This hypothesis is not met, it is not significant, so we cannot say that the perceived quality of municipal services by citizens living in urban areas positively and significantly modulates the direct relationship between the general perceived value of the city and citizen satisfaction.

Hypothesis 9 (Fig 11A) The perceived quality of the environmental experience of the resident citizen in an urban area directly and significantly moderates the causal relation

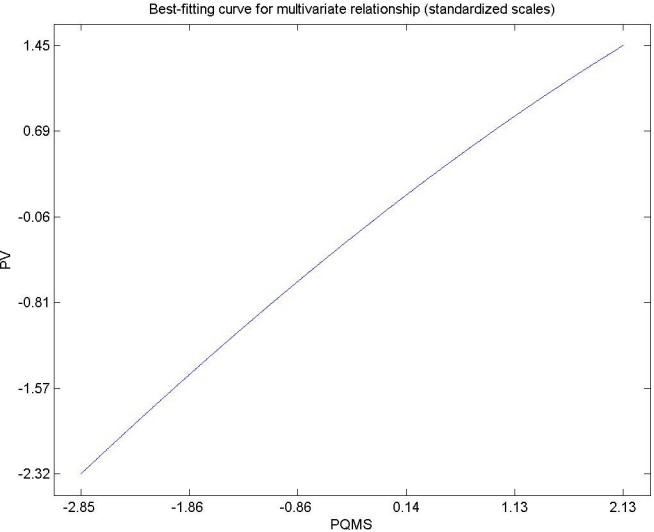

**Fig 7. Hypothesis 5, PQMS-PV.**

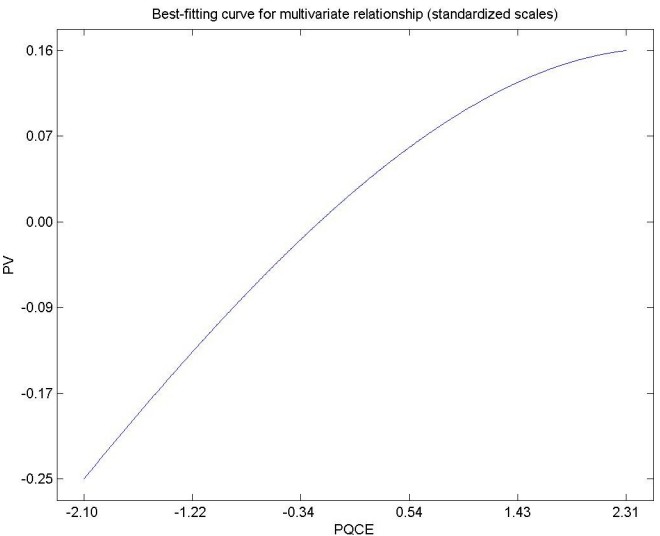

**Fig 8. Hypothesis 6, PQCE-PV.**

between perceived value and satisfaction [104, 105]. Fig 11B shows that when the values of the moderating variable (perceived quality in the environmental experience) are low, there is a direct causal relation between perceived value and satisfaction. When the moderating variable is high, the relationship between perceived value and satisfaction has a sinusoidal shape, flattened at the extremes, where the intensity of moderation of the environmental experience loses on the relationship between perceived value and satisfaction. The confirmation of this hypothesis means that by increasing the quality of the environmental experience of citizens living in urban areas, not only the perceived value is influenced but also the relationship between the perceived value and satisfaction, which means that increasing the quality of environmental services helps to increase the satisfaction and general well-being of the citizens residing in urban areas.

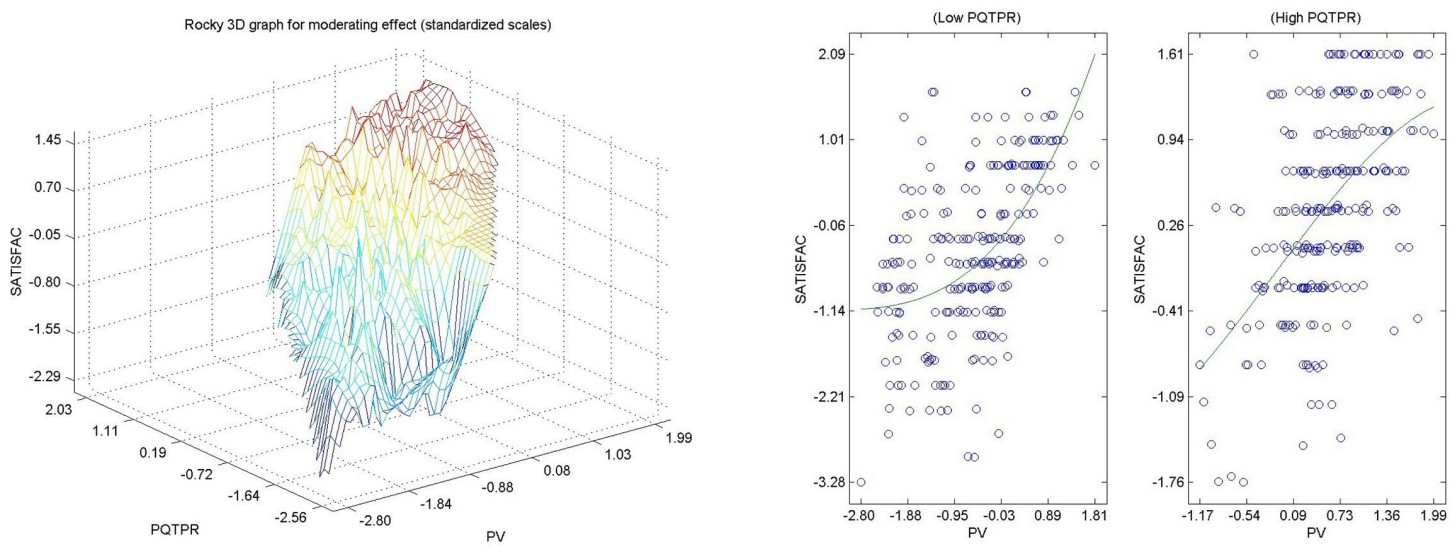

**Fig 9.** a. Hypothesis 7 3D, PQTRP->PV-SATISFAC. b. Hypothesis 7 2D, PQTRP->PV-SATISFAC.

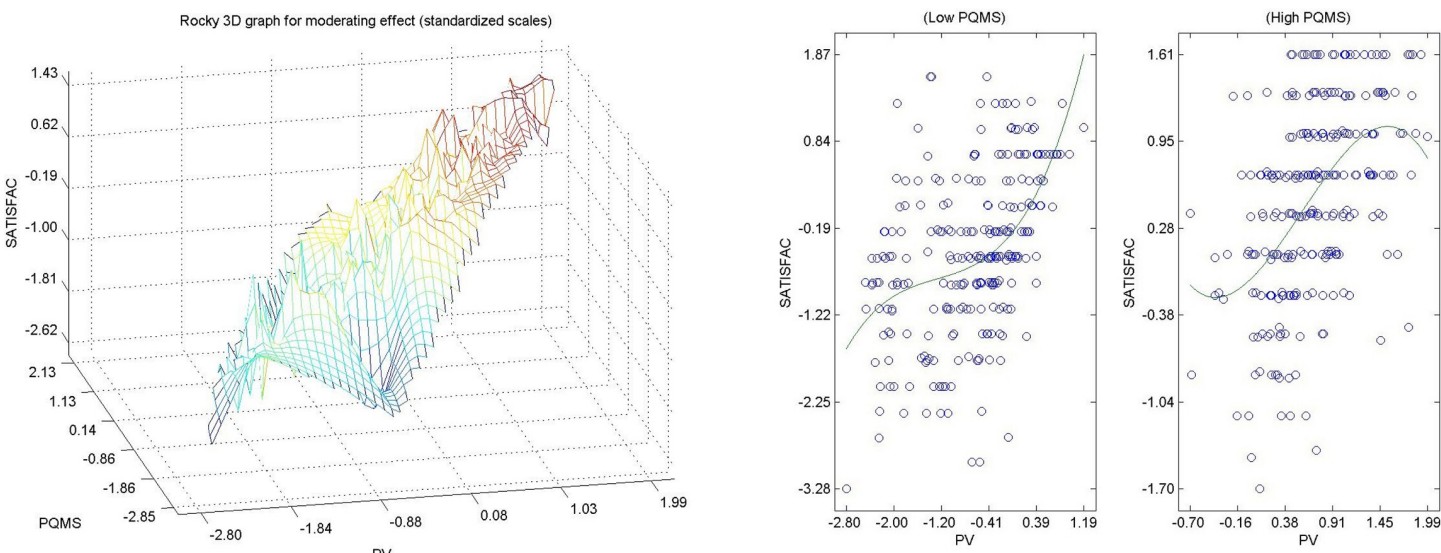

**Fig 10.** a. Hypothesis 8 3D, PQMS-> PV- SATISFAC. b. Hypothesis 8 2D, PQMS-> PV- SATISFAC.

Hypothesis 10 The value perceived by the resident citizen in an urban Area influences the satisfaction and well-being of the resident [106, 107]. Fig 12 shows a flattened sinus shape in the extreme right, in which the perceived value decreases its direct influence on the satisfaction of the citizen residing in urban Areas. This result has also been observed by other researchers. For the case under study, and for the expectations of the citizens of urban areas in general, the satisfaction of the residents can be increased by providing a set of experiences and sensations derived from territorial policies, viability, services public. and environmental orientation practiced by the municipality. This can be done by planning in an orderly manner in space in time a set of decisions and political measures that can generate the satisfaction of the resident in

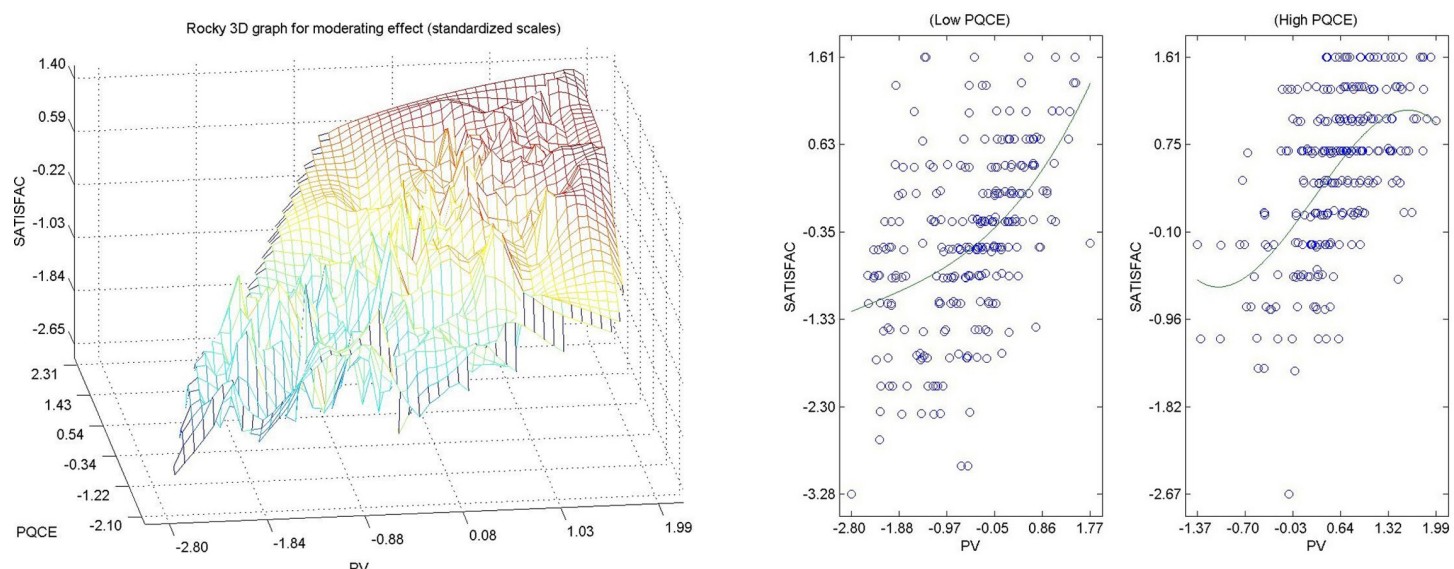

**Fig 11.** a. Hypothesis 9 3D, PQCE-> PV-SATISFAC. b. Hypothesis 9 2D, PQCE-> PV-SATISFAC.

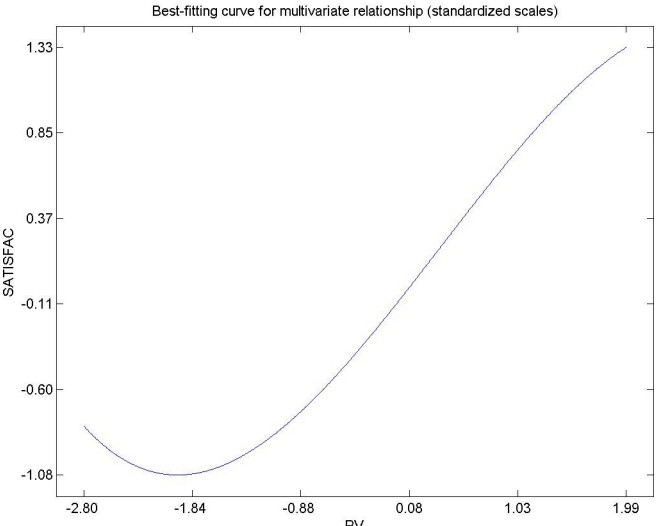

**Fig 12. Hypothesis 10, PV–SATISFAC.**

urban Areas, social well-being that fixes the population to the territory, avoiding the serious problem for the citizens. authorities of the territorial depopulation of these areas.

Hypothesis 11 was also confirmed, which shows the direct influence of citizen satisfaction in the urban areas in which they establish their residence on their loyalty, recommending relatives, friends and acquaintances, whether they reside in a rural area or in a different urban area [97, 108–110]. In this way, citizen loyalty not only fixes the population in the urban area, but also acts as a pole of attraction for other citizens to increase the population of the residential destination. Public managers have to find the right combination that maximizes public resources in the application of territorial policies, public and environmental services that maximize the satisfaction and loyalty of citizens in urban areas. Fig 13 shows the direct relationship of this relationship. The results show that citizen satisfaction, the quality of life and the expectations that each person has of their city, are positive factors that influence the decision-making of a citizen at the

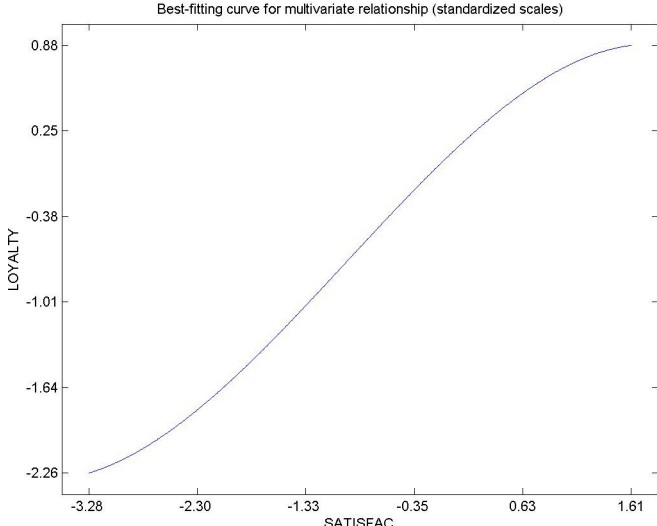

**Fig 13. Hypothesis 11, SATISFAC–LOYALTY.**

time of establishing their residence in an urban area, thus how to recommend said residential destination. This result is consistent with various studies of the literature. This study confirmed the relationship between high citizen satisfaction in the urban area of Guayaquil and the willingness of citizens to recommend Guayaquil as a residential destination. Deficiencies that affected the perceived quality in the municipal services area were also detected. These deficiencies included a lack of perceived quality in municipal police services. Citizen security is a fundamental factor that can attract or cause population movements. Likewise, the environmental experience of the citizen can be clearly improved in terms of air quality, noise pollution, green areas and recycling points. Deficiencies that affect the general perceived value of residents in Guayaquil were also found. These included the quality of the parks and gardens, adequate public lighting, and sports services. The satisfaction of the citizens was acceptable in general, although there were some irregularities to achieve a high fidelity levels.

This study outlines an achievable goal for public managers, who are in charge of territorial and viability planning, the municipal public services and the offer of public actions for an environmental experience of the citizen. If these elements improve, the experience of establishing their residence in an urban area can be maximized. This means that having satisfied urban citizens and well-being becomes one of the main actors in attracting and retaining people in urban areas. The results of this study are consistent with those found in previous researches that indicate that satisfaction positively influences the fidelity of citizens who establish their destination in an urban area and encourages them to recommend it. This study raises the most important factors to achieve the loyalty of the citizens of urban areas, generating positive synergies in the area, generating employment in various sectors, increasing demand by increasing the population that is installed in the urban destination.

This study identifies several factors that citizens living in urban areas consider important when determining their residence in an urban destination. Firefighting services were the most valued. Urban spaces must be capable of combining municipal public policies that link the citizen with the metropolis. The brand created by this urban destination, together with the perceived quality of the different municipal service areas, are factors that can be used to attract people who want to settle in these places.

The principal limitation of this study was related to the used sample with was obtained from the urban area of Guayaquil. It means this study is not generalizable to other populations. It would be desirable to study elsewhere where you can make comparisons of the results of the questionnaires. Another limitation is to carry out the study in a short period of time, and it would be convenient to do it periodically in order to analyze evolution.

## Author Contributions

**Conceptualization:** José Fernando Romero Subia.

**Data curation:** María Salomé Ochoa Rico.

**Formal analysis:** Arnaldo Vergara-Romero.

**Investigation:** Arnaldo Vergara-Romero, Juan Antonio Jimber del Río.

**Methodology:** Juan Antonio Jimber del Río.

**Project administration:** José Fernando Romero Subia.

**Resources:** José Fernando Romero Subia.

**Software:** Juan Antonio Jimber del Río.

**Validation:** Arnaldo Vergara-Romero.

**Writing – original draft:** María Salomé Ochoa Rico.

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
