## [Decision Letter · Decision Letter 0]

16 Nov 2021

PONE-D-21-28034Analysis of citizen satisfaction in urban areasPLOS ONE

Dear Dr. Vergara-Romero,

Thank you for submitting your manuscript to PLOS ONE. After careful consideration, we feel that it has merit but does not fully meet PLOS ONE’s publication criteria as it currently stands. Therefore, we invite you to submit a revised version of the manuscript that addresses the points raised during the review process.

Please consider all comments

We look forward to receiving your revised manuscript.

Kind regards,

Ahmed Mancy Mosa, Ph.D.

Academic Editor

PLOS ONE

2. Please consider changing the title so as to meet our title format requirement (https://journals.plos.org/plosone/s/submission-guidelines). In particular, the title should be "Specific, descriptive, concise, and comprehensible to readers outside the field" and in this case it is not informative and specific about your study's scope and methodology.

4. Thank you for submitting the above manuscript to PLOS ONE. During our internal evaluation of the manuscript, we found significant text overlap between your submission and the following previously published works, some of which you are an author.

- https://www.researchgate.net/publication/351365038_The_cultural_and_heritage_tourist_SEM_analysis_the_case_of_The_Citadel_of_the_Catholic_King

Please revise the manuscript to rephrase the duplicated text, cite your sources, and provide details as to how the current manuscript advances on previous work. Please note that further consideration is dependent on the submission of a manuscript that addresses these concerns about the overlap in text with published work.

We will carefully review your manuscript upon resubmission, so please ensure that your revision is thorough

Reviewers' comments:

Reviewer's Responses to Questions

**Comments to the Author**

1. Is the manuscript technically sound, and do the data support the conclusions?

Reviewer #1: Yes

Reviewer #2: Yes

2. Has the statistical analysis been performed appropriately and rigorously? 

Reviewer #1: Yes

Reviewer #2: Yes

3. Have the authors made all data underlying the findings in their manuscript fully available?

Reviewer #1: Yes

Reviewer #2: Yes

4. Is the manuscript presented in an intelligible fashion and written in standard English?

Reviewer #1: Yes

Reviewer #2: Yes

5. Review Comments to the Author

Reviewer #1: Dear authors,

I read with interest your paper, although, after assuring the p-value and VIF were ok, I slightly "jumped" here and there the more technical parts trusting your academic reliability.

I like the differences you mentioned between expected and perceived and I thought that some literature within this recent publication https://www.sciencedirect.com/science/article/pii/S1877916621000059 would be of interest for your paper, especially for the introduction and conclusion when linking with the state of the art.

Your paper appears to me well structured even if some improvements in the theoretical parts related with quality of life in general, and the urban quality of life in particular would be beneficial.

Reviewer #2: It is an interesting paper that focuses on the citizens’ satisfaction, the expected quality, and perceived quality. The method and results show credible. However, there are some questions that I think should be stated again.

1. How is the state of arts of your research issue? There is no related works review.

2. We have much research about citizens’ perception and satisfaction with urban management. What are your contributions, especially the theoretical contributions? Though you have shown in the section of the introduction, what’s the difference between yours and others?

3. What’s the meaning of ‘experience in territorial and road planning’? Is it pointed to someone who has participated in the territorial and road planning of the urban?

4. The part of ‘Multigroup Analysis’ seems no direct relationship with your research. While you just show the multigroup in gender, and how are other groups (education? Work? Income?) So, maybe you can give up this part.

6. PLOS authors have the option to publish the peer review history of their article (what does this mean?). If published, this will include your full peer review and any attached files.

Reviewer #1: No

Reviewer #2: No

---

## [Author Response · Author response to Decision Letter 0]

9 Dec 2021

Revision required [PONE-D-21-28034] - [EMID:847aab6a5cba99f6]

2. Please consider changing the title so as to meet our title format requirement (https://journals.plos.org/plosone/s/submission-guidelines). In particular, the title should be "Specific, descriptive, concise, and comprehensible to readers outside the field" and in this case it is not informative and specific about your study's scope and methodology.

4. Thank you for submitting the above manuscript to PLOS ONE. During our internal evaluation of the manuscript, we found significant text overlap between your submission and the following previously published works, some of which you are an author.

Please revise the manuscript to rephrase the duplicated text, cite your sources, and provide details as to how the current manuscript advances on previous work. Please note that further consideration is dependent on the submission of a manuscript that addresses these concerns about the overlap in text with published work.

We will carefully review your manuscript upon resubmission, so please ensure that your revision is thorough

Dear editor and reviewers, thank you very much for allowing us revising and resubmitting our article to the PLOS ONE. We have found your comments to be highly helpful in improving the article in terms of the introduction, the theoretical background, the research methodology, and the theoretical contributions. After carefully reading your comments, we have introduced some changes in the manuscript to address your concerns. Following your recommendation, as shown in the new version of the manuscript. We present our detailed comments below.

Following the editor's recommendations, we have proceeded to rewrite paragraphs that were similar to previous works by the authors, these changes can be checked in "Revised Manuscript with Track Changes"

The new title is STUDY OF CITIZEN SATISFACTION AND LOYALTY IN THE URBAN AREA OF GUAYAQUIL: PERSPECTIVE OF THE QUALITY OF PUBLIC SERVICES APPLYING STRUCTURAL EQUATIONS.

After following the editor's recommendations, we detail the changes made by following the reviewers' comments

Reviewers

1. Is the manuscript technically sound, and do the data support the conclusions?

Reviewer #1: Yes

Reviewer #2: Yes

Thank you very much for your constructive suggestion

2. Has the statistical analysis been performed appropriately and rigorously?

Reviewer #1: Yes

Reviewer #2: Yes

Thank you very much for the review. The authors agree in their comments in particular, we greatly appreciate their contributions to improve the manuscript.

3. Have the authors made all data underlying the findings in their manuscript fully available?

Reviewer #1: Yes

Reviewer #2: Yes

Thank you very much

4. Is the manuscript presented in an intelligible fashion and written in standard English?

Reviewer #1: Yes

Reviewer #2: Yes

Thank you very much

5. Review Comments to the Author

Reviewer #1: Dear authors,

I read with interest your paper, although, after assuring the p-value and VIF were ok, I slightly "jumped" here and there the more technical parts trusting your academic reliability.

I like the differences you mentioned between expected and perceived and I thought that some literature within this recent publication https://www.sciencedirect.com/science/article/pii/S1877916621000059 would be of interest for your paper, especially for the introduction and conclusion when linking with the state of the art.

Your paper appears to me well structured even if some improvements in the theoretical parts related with quality of life in general, and the urban quality of life in particular would be beneficial.

Thank you very much for your constructive suggestion. The suggestion to include references to the article Preferring or Needing Cities? (Evolutionary) psychology, utility and life satisfaction of urban living. It is really interesting and improves our manuscript.

The following paragraphs have been introduced in the introduction and in the conclusions:

Introduction: 

For a citizen to want to establish his residence in a metropolis, he considers the quality of life in it. The literature indicates that the larger the cities, the lower the level of satisfaction, due to insecurity and stress. Despite this, most people live in cities, due to the public services they offer in terms of quality and quantity. Rural areas are neglected in terms of basic services, their citizens having to move to the cities to satisfy medical, leisure, specialized supplies, etc [21].

[21] L. S. D’Acci, “Preferring or Needing Cities? (Evolutionary) psychology, utility and life satisfaction of urban living,” City, Culture and Society, vol. 24, 2021, doi: 10.1016/j.ccs.2021.100375.

The literature indicates that in rich countries rural life is preferred, while in poor or developing countries life in urban areas is preferred [22]. Citizens have the belief that all these wonderful things (money, creativity ...) attract them to the big cities without taking into account negative aspects (crime, stress, congestion, pollution, lack of nature,….) [23].

[22] M. das Gupta, J. Zhenghua, L. Bohua, X. Zhenming, W. Chung, and B. Ha-Ok, “Why is son preference so persistent in East and South Asia? A cross-country study of China, India and the Republic of Korea,” The Journal of Development Studies, vol. 40, no. 2, pp. 153–187, 2003.

[23] Kahneman, D., Kahneman, D., & Tversky, A. (2000). Experienced utility and objective happiness: A moment-based approach. 2000, 673-692.

Conclusions: 

There are various reasons why citizens decide to live in metropolises; despite sacrificing uses related to happiness and quality of life, such as air quality and environmental quality, to achieve other advantages such as social mobility, money, culture and professional satisfaction [21].

[21] L. S. D’Acci, “Preferring or Needing Cities? (Evolutionary) psychology, utility and life satisfaction of urban living,” City, Culture and Society, vol. 24, 2021, doi: 10.1016/j.ccs.2021.100375.

Reviewer #2: It is an interesting paper that focuses on the citizens’ satisfaction, the expected quality, and perceived quality. The method and results show credible. However, there are some questions that I think should be stated again.

1. How is the state of arts of your research issue? There is no related works review.

The state of the art of this research focuses on the importance of the adequate provision of public services in urban areas analyzed from the causal relationship between expected quality and perceived quality, whose difference establishes a perceived value, in order to influence the level of citizen satisfaction, which generates loyalty or rootedness to the territory, as well as improvement in the quality of life. 

Thank you for this important remark. Next, we add reviews of related works.

[7] S. P. M. Pereira and P. M. A. R. Correia, “Sustainability of portuguese courts: Citizen satisfaction and loyalty as key factors,” Sustainability (Switzerland), vol. 12, no. 23, 2020, doi: 10.3390/su122310163.

[8] I. Almarashdeh, “The effect of recovery satisfaction on citizens loyalty perception: A case study of mobile government services,” International Journal of Electrical and Computer Engineering, vol. 10, no. 4, 2020, doi: 10.11591/ijece.v10i4.pp4279-4295.

[9] A. Alkraiji and N. Ameen, “The impact of service quality, trust and satisfaction on young citizen loyalty towards government e-services,” Information Technology and People, 2021, doi: 10.1108/ITP-04-2020-0229.

[16] X. mei Fu, J. hua Zhang, and F. T. S. Chan, “Determinants of loyalty to public transit: A model integrating Satisfaction-Loyalty Theory and Expectation-Confirmation Theory,” Transportation Research Part A: Policy and Practice, vol. 113, 2018, doi: 10.1016/j.tra.2018.05.012.

[24] L. Ma, “Performance Management and Citizen Satisfaction With the Government: Evidence From Chinese Municipalities,” Public Administration, vol. 95, no. 1, pp. 39–59, 2017, doi: 10.1111/padm.12275.

[32] S. Parnell and E. Pieterse, “The ‘right to the city’: Institutional imperatives of a developmental state,” International Journal of Urban and Regional Research, vol. 34, no. 1, pp. 146–162, 2010, doi: 10.1111/j.1468-2427.2010.00954.x.

[50] G. van Ryzin, “The Measurement of Overall Citizen Satisfaction,” Public Performance & Management Review, vol. 27, no. 3, 2004, doi: 10.1080/15309576.2004.11051805.

[51] M. H. Tammubua, “ANALISA PENGARUH SELF IMAGE CONGRUITY, RETAIL SERVICE QUALITY, DAN CUSTOMER PERCEIVED SERVICE QUALITY TERHADAP CUSTOMER LOYALTY YANG DIMEDIASI CUSTOMER SATISFACTION URBAN SURF/DISTRO DI JAYAPURA,” Jurnal Organisasi dan Manajemen, vol. 13, no. 2, 2017, doi: 10.33830/jom.v13i2.68.2017.

2. We have much research about citizens’ perception and satisfaction with urban management. What are your contributions, especially the theoretical contributions? Though you have shown in the section of the introduction, what’s the difference between yours and others?

Thank you for your comment. Although it is true that there are various studies of citizen satisfaction, most of them start from expectations and perceived quality in general, we have segmented into three dimensions this analysis of causal relationships between Expected Quality and Perceived Quality, which are: territorial planning and roads, Municipal services and environmental management in order to determine the level of satisfaction and loyalty independently based on constructs formed by items of the analyzed dimension, which show different coefficients between dimensions, which can be comparable and even generate future research. In this way, specific tools can be made available to public managers to improve the quality of life of citizens.

Also, we propose an analysis based on moderating variables of the proposed causal relationships in order to know if there is a degree of influence by dimension of the constructs of perceived quality within the causal relationship that determines the degree of satisfaction.

The results of this research by dimension generate theoretical contributions related to the importance of measuring the level of satisfaction and rootedness for each area that makes up public management, whose contributions will also be included in the conclusions as you rightly indicate.

3. What’s the meaning of ‘experience in territorial and road planning’? Is it pointed to someone who has participated in the territorial and road planning of the urban?

Thank you for your comment, theexperience in territorial and road planning, is the general result of the construct formed by a set of items that address issues related to the level of satisfaction in the provision of services inherent to territorial planning and roads. Some of the items that make up this construct are: Zoning and urban planning, Roads and pavements, Traffic organization, Public Transport Service, Parking services, Address information, Transport terminal services. The results of the set of items proposed for this construct in the citizen surveys generate a degree of general experience on the part of the citizens surveyed.

4. The part of ‘Multigroup Analysis’ seems no direct relationship with your research. While you just show the multigroup in gender, and how are other groups (education? Work? Income?) So, maybe you can give up this part.

Thank you very much for your support. In the multigroup analysis, we expected to find more significant differences between men and women.

Following your recommendation, we have proceeded to delete the multi-group analysis paragraph.

---

## [Editor Report · Decision Letter 1]

17 Jan 2022

Study of citizen satisfaction and loyalty in the urban area of guayaquil: perspective of the quality of public services applying structural equations

PONE-D-21-28034R1

Dear Dr. Vergara-Romero,

We’re pleased to inform you that your manuscript has been judged scientifically suitable for publication and will be formally accepted for publication once it meets all outstanding technical requirements.

Kind regards,

Ahmed Mancy Mosa, Ph.D.

Academic Editor

PLOS ONE
---

## [Editor Report · Acceptance letter]

26 Jan 2022

PONE-D-21-28034R1 

Study of citizen satisfaction and loyalty in the urban area of Guayaquil: perspective of the quality of public services applying structural equations 

Dear Dr. Vergara-Romero:

I'm pleased to inform you that your manuscript has been deemed suitable for publication in PLOS ONE. Congratulations! Your manuscript is now with our production department. 

Kind regards, 

on behalf of

Dr. Ahmed Mancy Mosa 

Academic Editor

PLOS ONE